# Sustainable Soil and Water Resources Management in Nigeria: The Need for a Data-Driven Policy Approach

**Kennedy O. Doro** [1,*] **, Solomon Ehosioke** [2] **and Ahzegbobor P. Aizebeokhai** [3]

1  Department of Environmental Sciences, University of Toledo, 2801 West Bankcroft St., Toledo, OH 43606, USA

2  Department of Urban and Environmental Engineering, University of Liège, B 400 Liège, Belgium; solomon.ehosioke@uliege.be

3  Department of Physics, Covenant University, Ota P.M.B. 1023, Nigeria; philips.aizebeokhai@covenantuniversity.edu.ng

*  Correspondence: kennedy.doro@utoledo.edu; Tel.: +1-419-530-2811

**Abstract:** Effective public policies are needed to manage a nation's natural resources, including soil and water. However, making such policies currently requires a shift from a traditional qualitative approach to a mix of scientific data, evidence and the relevant social elements, termed data-driven policymaking. Nigeria, like most developing countries, falls short of the framework for this approach. Nevertheless, the lack of potable water in some regions and the continuous degradation of farmable lands call for intervention through effective policy formulation and implementation. In this work, we present a conceptual workflow as a strategic step towards developing a framework for a data-driven soil and water resources management policy. A review of the current legal and policy framework and selected scientific literature on soil and water resources in Nigeria is presented. Analysis of the National Water Resources Bill proposed in 2018 is used to highlight existing gaps between policy, scientific data and reality. Modern field techniques and project-based examples for soil and aquifer characterization that can be adapted for local use are presented. While government must take responsibility for the poor policy framework, the research community is challenged on the need for scientific data as a base for effective policy formulation and implementation.

**Keywords:** policy; data-driven; sustainability; soil and water resources; Nigeria

## 1. Introduction

There is a global consensus among scientists and policy experts, at least in recent times, on the need to formulate policies that address access to and protection of basic environmental resources including soil and water [1–5]. Decades of intense agricultural practices, mining, wars, and industrial growth have left behind legacies of soil and water pollution. This, coupled with the recent impacts of global climate change, limits availability of suitable soil for agriculture and access to clean water for domestic and industrial uses in some part of the world, including sub-Saharan Africa [6–9]. Hence, there is a growing need for an urgent, effective, and sustainable management strategy for soil, water, and the environment in general. This study focusses on the need for a data-driven policy approach for sustainable management of soil and water resources in Nigeria.

The effective and sustainable management of soil and water resources, as in many other areas, would require the right policy framework, with appropriate laws and regulations and a willingness to implement them. Such a policy framework must transcend previous non-empirical approaches, which have relied mainly on limited personal or group experiences, opinions, instincts, dogma, beliefs, etc., to a quantitative approach that uses a mix of high-quality scientific data, calibrated models

and evidence with the relevant societal and social elements [10]. This approach, which is becoming increasingly popular in governance, policing, transportation and the corporate business world, has been described by various authors as data-driven policy or decision making [11,12]. With the current level of advancement in software, instrumentation, computational power and techniques, the possibility of acquiring and processing large volumes of data on most aspects of human endeavor has increased greatly. The challenge is gradually shifting from data availability to effective use of available data to improve society and mankind. This, basically, is the driving concept of data-driven policymaking, which challenges policymakers, mostly government, to be more pragmatic and efficient in policy formulation and implementation. For the case of soil and water resources management, applying such data-driven policy strategy will require a scientific database containing the appropriate qualitative and quantitative datasets describing the state as well as the physical and biogeochemical processes occurring in soil and water systems.

In the last decades, several advances have been made in numerical, laboratory and field studies in characterizing, monitoring and remediating soils and surface and groundwater systems [6,13–15]. Particularly for soil and groundwater systems which are associated with a highly heterogeneous shallow subsurface, several numerical and experimental techniques have been developed to better characterize the soil and aquifer architecture, monitor their fluid and biogeochemical dynamics, and, as well, estimate the necessary parameters controlling the flow and transport of water and other biogeochemical constituents through them. Although this has remained a major challenge to both researchers and practitioners of hydrogeological, geophysical, biogeochemical and other related research [16–21], the advances to date provide an opportunity for a more efficient management of our soils and water resources than we currently do through a data-driven policy approach.

The concept of data-driven policy, despite its enormous potential, assumes the availability or at least the capacity for making the required data available as well as analyzing them. This, considering the level of scientific advancement as well as the present legal and public policy framework in Nigeria and most developing countries, presents a dual challenge. The first is that Nigeria, like most other sub-Saharan African countries, falls short in the legal and public policy framework for a pragmatic formulation and implementation of laws and regulations for effective management of the nation's soils and water resources [22–24]. Although soil, water and other environment-related legislations exist in Nigeria, they can best be described as non-pragmatic [9,23] as they have had minimal impact on the issues of resource ownership, usage, protection, liability and responsibility for remediation in cases of contamination [9,22–25]. Existing legislations, such as the Land Use Act of 1978, River Basin Development Act of 1986 amended in 2017, Federal Environmental Protection Agency Act of 1988, Decree 101 of 1990, Water Resources Decree of 1993, National Environmental Standards and Enforcements Agency Act of 2007, etc., have not had the desired impact [24,25]. This is not only a result of lack of enforcement but also that they were based, ab initio, primarily on socio-economic factors, and lack the necessary scientific framework, such as a detailed assessment of the resource vulnerability and sensitivity [23,25,26]. The second challenge, which is also critical, is the availability of the required database to drive the needed policies. There have been limited experimental and detailed numerical studies on the estimation of hydraulic, physical and biogeochemical parameters required for understanding and managing soil and water resources in the country. Hence, the database to drive informed policy is practically missing. Where data exist, they are uncoordinated and poorly managed.

In view of the above, this study reviews selected literature on soil and water resources in Nigeria to give an overview of the knowledge base and state of the art locally in comparison with the global state of the art, with emphasis on characterization, monitoring and parameter estimation techniques. An insight into the major policies and laws on water resources in Nigeria, with a review of the National Water Resources Bill proposed in 2018 but stepped down by parliament for a more detailed review, is presented to highlight the gap between policies, scientific data and reality. An overview of the development and field application of selected classical and modern, cost effective and innovative techniques for soil and aquifer characterization is also presented. Emphasis is, however, on low-cost

techniques with great potential for implementation in Nigeria considering a possible low budget. This could encourage the needed scientific exchange and adaptation, and potentially contribute to advancing the state of the art as well as the conducting of field experiments to create a soil and groundwater parameters database. A brief comparative insight into the policy framework and implementation for protecting and managing soil and water resources in other developed countries and Nigeria is also highlighted, with possible lessons that can be learnt and adapted in Nigeria and other developing countries with similar situations. This work concludes by challenging the current policy framework for soil and water resource protection and management in Nigeria, presenting a conceptual workflow as a strategic step towards developing a framework for a data-driven policy approach and challenging the scientific community on its role in providing the needed data and models to support the desired policy framework.

## 2. An Overview of Nigeria

Nigeria is situated in the West African sub-region on the Gulf of Guinea and has a total landmass of 923,768 sq. km (Figure 1). It lies between latitudes 4° and 14° N, and longitudes 3° and 14° E. Nigeria is bordered to the north by the Republics of Niger and Chad, and to the west by the Republic of Benin. It shares its eastern borders with the Republic of Cameroon down to the shores of the Atlantic Ocean, which forms the southern limit of its territory. Its coastline is about 853 km and possess abundant arable land for agricultural, industrial and commercial activities.

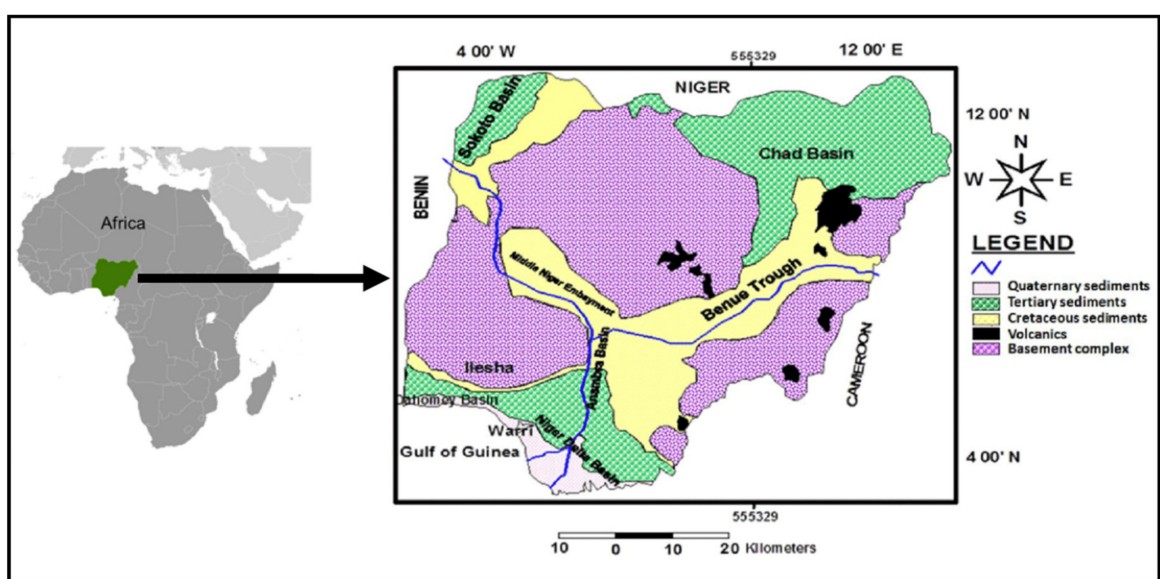

**Figure 1.** Map of Nigeria with its border countries (**right**) and its location in the African continent (**left**).

Present-day Nigeria comprises of 36 states and the Federal Capital Territory located in Abuja. The constitution of the Federal Republic of Nigeria defines the country as a democratic secular state. Nigeria has been home to several ancient and indigenous kingdoms and states over the millennia. The present-day territorial shape of Nigeria originated from British colonial rule in the 19th century due to the amalgamation of the Southern and Northern Nigeria Protectorate in 1914 by Lord Fredrick Lugard. The British established administrative and legal structures while practicing indirect rule through traditional chiefdoms [27]. Nigeria formally became an independent federation in 1960 but experienced a civil war between 1967 and 1970. Thereafter, the Nigerian nation alternated between democratically elected civilian governments and military dictatorships until it eventually achieved a stable multi-party democracy in 1999. Nigeria's economy is predominantly driven by oil revenues from the Niger Delta fields; however, about one third of Nigerians are employed in the agriculture sector. The service sector is large, particularly telecommunications and financial services, and the

manufacturing industry is growing, with lots of potential. Nigeria has extensive solid mineral resources, but the mining industry is still underdeveloped.

Nigeria is within the tropics, but its climate varies from tropical in the south to semi-arid in the north. The climate is marked by two seasons, rainy and dry seasons, with a short dry period referred to as August break, which lasts two to three weeks in August. The rainy season spans between April and October, while the dry season is from November to March. Monthly temperatures in the coastal areas of the south range from 21 °C to 37 °C, while extreme temperatures in the north, with a much drier climate, range from 40° to 50 °C. Mean annual rainfall is about 2300 mm in the south, with most of the precipitation received during the rainy season. Nigeria has extensive surface water resources; the main rivers Niger and Benue converge and flow together into the Atlantic Ocean through the Niger Delta (Figure 1). Groundwater is widely used for domestic, agricultural and industrial supplies. Most rural areas and a number of towns and cities are dependent on groundwater.

## 3. State of the Art and Review of Soil and Groundwater Research in Nigeria

The shallow subsurface, which includes the soil, vadose zone and aquifers usually up to a depth of a few tens to hundred meters, is characterized by a high level of heterogeneity with its properties showing high spatial variability. This has direct consequences for the management of shallow aquifers used for the provision of drinking water, irrigation, and other purposes as well as soils for agriculture. The spatial variability of subsurface parameters, such as porosity, hydraulic conductivity, moisture content, etc., causes groundwater flow and other transport processes to be quite non-uniform [28]. This invariably impedes assessment and exploration for groundwater, design of sustainable irrigation systems, optimal use of soil for agricultural purposes as well as the remediation of contaminated aquifers and soils. The direct consequences of this are sub-optimal use of groundwater resources, uncertainties in the design of water supply and irrigation systems and a high level of risk in the design of protective measures for soils and aquifers [29,30]. Therefore, a comprehensive assessment of the hydraulic as well as other physical and biogeochemical properties of soils and aquifers and their spatial variability is necessary for the effective management and protection of soils and groundwater resources.

Several methods have been developed for characterizing variability in hydraulic and biogeochemical parameters of soils and aquifers. For soil moisture characterization, in-situ measurement is possible using commonly available probes. However, recent approaches involve integrating remote sensing or geophysical measurement with such in-situ measurement for large-scale characterization [31]. Aquifer characterization generally requires conducting hydraulic tests such as pumping and tracer tests for estimating spatially averaged flow and transport parameters such as hydraulic conductivity, porosity and dispersivity, to name the most important [17,32]. However, such tests inherently fail to resolve detailed heterogeneity. Other techniques, such as multilevel slug tests, flowmeter measurements and direct-push-based methods, can resolve the variation of hydraulic conductivity at high vertical resolution [33–36]. These techniques give point information and require a high number of measurements to characterize three-dimensional heterogeneity within a small domain. To increase the spatial resolution of hydrogeological investigations, geophysical techniques may be combined with hydrogeological methods [37]. New techniques such as hydraulic and tracer tomography have also been developed for high-resolution aquifer characterization [13,15,16,38,39]. For relating geophysical data to soil moisture, petrophysical relationships are often explored [40]. While such petrophysical relationships are also useful for relating geophysical data to aquifer hydraulic parameters, current approaches involve the use of fully coupled hydrogeophysical inversion techniques [38,41]. Additionally, estimating hydraulic parameters such as hydraulic conductivity from aquifer tests requires inverse modeling and parameter estimation techniques, which have been highly studied as well [28,42].

The growing concerns about sustainable use of soil and water resources have also attracted extensive research in Nigeria within the past few decades. Most of these researches focused on delineating aquifer architecture, geochemical characterization of soils, hydrogeochemical studies of

surface and groundwater quality, and environmental impact assessment, possibly in response to increasing threats of contamination resulting from landfills, indiscriminate waste disposal and use of fertilizers, hydrocarbons and other sources [9,23,43–46]. Research characterizing soil moisture and aquifer heterogeneities using hydrogeological and geophysical techniques [47–49] as well as groundwater flow and contaminant transport modeling [50–52] has also received some attention. However, detailed numerical and experimental research on the quantification of spatially varying soil and aquifer hydraulic parameters necessary to reduce uncertainties in engineering designs for irrigation and groundwater extraction, vulnerability assessment, and protection and remediation of soil and groundwater resources has received little attention [23]. A reason for this is often the lack of access to modern equipment required for conducting field-scale experiments outside the oil-and-gas sector. This may also be connected to lack of attention to or inexperience in adapting current state of the art technology for use in Nigeria using low-cost sensors and equipment. With the increasing trend in surface and groundwater contamination in Nigeria as well as the need for effective soil management for improved agricultural yield, there is a pressing need to develop locally sustainable technology and formulate policies for optimum management of soil and groundwater resources.

## 4. Soil and Water Resources Related Policies in Nigeria

Several laws, regulations and policies have evolved in Nigeria post-independence with the aim of addressing issues related to soil and water resource ownership, exploitation, protection and management. In the following section, we present an overview of some major soil- and water-resource-related regulations in Nigeria. In assessing these existing policies, we evaluated their success as a measure of their effectiveness at achieving their set objectives. This approach follows the logic model of policy analysis relying on public perception and existing reality in assessing policy effectiveness [53]. This approach has been recommended as an option of last resort when little or no data are available for evaluating policy effectiveness [53]. We acknowledge the potential bias with this approach due to the difficulty in establishing the cause-and-effect relationship and the fact that public policies are just one of the factors that could simultaneously affect the targeted problem [54]. However, for the Nigerian scenario with very limited data and detailed prior studies, this could provide a base for further studies. While such study is beyond the scope of this paper, our focus is to stimulate a positive interaction between science and policy for improving current soil and water resources management policies in the country.

Prior to 1978, the ownership and use of land, including access to its resources (water and soil for farming), was controlled by the land tenure system involving individual, group and communal ownership, mainly governed by the customary law [55]. The Land Use Act of 1978, however, vested the right of ownership of all lands in the territory of each State, except land belonging to the Federal government or its agencies, solely in the Governor of the State. The Governor holds such land in trust for the people and is responsible for allocation of land in all urban areas to individuals and organizations for residential, agricultural, commercial and other purposes. Similar powers with respect to non-urban areas were conferred on Local Governments [56]. Both the customary-law-based land tenure system and the Land Use Act only address ownership vaguely, resulting in multiple litigations [57,58]. While these laws are often referred to on issues relating to soil and water resources in Nigeria, they do not address specific issues relating to access, protection, liability, remediation and management of soil and water resources on these lands.

A major act of parliament in Nigeria with a direct focus on water resource management is the River Basin Development Act. The Act was first enacted in 1979, modified in 1986 and last amended in 2017. The act, in its current state, establishes thirteen River Basin Development Authorities in Nigeria (see Table 1). The responsibilities of each authority include developing both surface and groundwater resources, with emphasis on irrigation, flood and erosion control and watershed management; supplying water to users at a pre-approved fee; and maintaining a comprehensive water resources master plan with all water resource requirements in the Authority's area of operation,

through adequate collection and collation of the relevant data on the River Basin. The outlined objectives of the River Basin Development Authorities, if well implemented, would provide an effective framework for soil and water resource management across the country on a basin scale. However, the river basin development authorities fall short in most of these duties individually, with over 30% of Nigerians lacking access to clean drinking water [59], while seasonal drought, erosion and flooding remain a major challenge for the country [60]. Additionally, the lack of collaborative efforts across basin authorities make surface and groundwater management both at small and catchment scales practically ineffective [53]. The responsibilities of the authorities require a scientific database for effective implementation. Knowledge of soil and aquifer heterogeneities and surface and groundwater hydraulic properties is needed for managing these resources, both locally and across catchments. These datasets and detailed scientific studies are missing, which could account for the suboptimal performance of the Authorities' irrigation, flood management, water supply and other related projects [9,23].

**Table 1.** List of River Basin Development Authorities in Nigeria.

| S/N | River Basin Authority | States Covered | Head Office |
|---|---|---|---|
| 1. | Anambra River Basin Authority | Anambra, Enugu and Ebonyi states | Enugu |
| 2. | Benin-Owena River Basin Development Authority | Edo, Ekiti and Ondo states and areas of Delta State drained by Benin, Escravos, Forcades and Ramos river creek systems | Benin |
| 3. | Chad Basin Development Authority | Bornu and Yobe states and areas of Adamawa State drained by the Yedseram and Goma river systems | Maiduguri |
| 4. | Cross River Basin Development Authority | Akwa Ibom and Cross River states | Calabar |
| 5. | Hadejia-Jama'are River Basin Development Authority | Jigawa and Kano states and areas of Bauchi state drained by the Misau and Jama'are river systems | Kano |
| 6. | Imo River Basin Development Authority | Abia and Imo states | Owerri |
| 7. | Lower Benue River Basin Development Authority | Benue, Nasarawa and Plateau states | Makurdi |
| 8. | Lower Niger River Basin Development Authority | Kwara and Kogi states | Ilorin |
| 9. | Niger Delta River Basin Development Authority | Rivers, Bayelsa and part of Delta state | Port-Harcourt |
| 10. | Ogun-Osun River Basin Development Authority | Lagos, Ogun, Osun and Oyo states | Abeokuta |
| 11. | Sokoto-Rima River Basin Development Authority | Kastina, Kebbi, Sokoto and Zamfara states | Sokoto |
| 12. | Upper Benue River Basin Development Authority | Gombe and Taraba states and part of Bauchi State drained by the Gongola River system and the whole of Adamawa, excluding the area drained by the Yedseram River system | Yola |
| 13. | Upper Niger River Basin Development Authority | Niger and Kaduna states and the Federal Capital Territory | Minna |

The Federal Environmental Protection Agency (FEPA) Act of 1988, amended in 1992, established the agency with the responsibility for protecting and developing the environment in general and environmental technology, including initiation of policy in relation to environmental research and technology. The agency's duties include:

(a)　Advising the federal government on national environmental policies and priorities and on scientific and technological activities relating to the environment.

(b)　Preparing periodic master plans for the development of environmental science and technology and advising the federal government on their financial implications.

(c)　Promoting co-operation with similar local and international organizations in environmental science and technology connected with the protection of the environment.

(d)　Co-operating with federal and state ministries, local government councils, statutory bodies and research agencies on matters and facilities relating to environmental protection.

(e)　Carrying out other activities that are considered necessary or expedient for the full discharge of the functions of the agency outlined in the Act.

The Federal Environmental Protection Agency, just like the river basin development authorities with framework replicated in different states and local councils across the country, has lofty objectives. However, these institutions lack the structures and support to achieve their set-out objective, which is a major reason accounting for the poor management of soil, water and other environmental resources across the country [61].

With a legacy of military rule lasting about three decades in Nigeria, the military promulgated decrees, some of which were aimed at managing the Nation's soil and water resources. One of such decree is the Water Resources Decree of 1993, also referred to as decree 101. The decree, like the Land Use Act of 1978, vested the right to use and control of both surface and groundwater cutting across more than one state as well as associated riverbeds and banks on the federal government. The decree also did not result in any significant change in the access to and protection of soil and water resources in the country [55].

An act of parliament in Nigeria that focused on environmental standards and regulation is the National Environmental Standards and Enforcements Agency Act of 2007. The act established the agency with the responsibility to protect and develop the environment, conserve its biodiversity and ensure sustainable development of Nigeria's natural resources and environmental technology, including coordination and liaison with relevant stakeholders within and outside Nigeria on matters of enforcement of environmental standards, regulations, rules, laws, policies and guidelines within the country. While the agency is saddled with the responsibility of implementing environmental standards, these standards are, however, not well defined, and in cases where they are, such standards are neither based on nor updated by a scientific database [62,63].

Nigeria also has a Hydrological Services Agency (NIHSA) which is an arm of the Federal Ministry of Water Resources. It was established by an act of parliament in 2010, with a major objective to advise the federal and state government on water-related policies and, as well, prepare, project and interpret such policies. The agency provides surface and groundwater resource assessment in terms of quantity, quality, distribution and availability in time and space, for efficient and sustainable management. It is expected to operate and maintain hydrological stations across the country and carry out groundwater exploration and monitoring to provide the needed hydrological data for planning, design, execution and management of water resources and allied projects. Several other acts of parliament, agencies and institutions exist with the aim and responsibility of managing soil and water resources in Nigeria. Some of these include the Water Resources Act of 2004, Federal Ministry of Water Resources, Nigerian Integrated Water Resources Management Commission and the National Water Resource Institute.

From the above review, it is obvious that the approach of the Nigerian government over the years towards managing the nation's soil and water resources has been largely uncoordinated, resulting in multiple laws and agencies with duplicated functions and inefficiency that make it difficult to really achieve the set-out goal. Apart from the duplication of functions, the established agencies lack consistent funding and independence with the appropriate checks to keep and improve on their activities irrespective of changes in government. This reflects a lack of political will and insincerity on the part of Government in the establishment of these agencies. Additionally, consistent with almost all the laws and agencies reviewed is the fact that their enactment and establishment is driven only by

social and political interests and lacks the scientific base to address the challenges they are established for. Hence, to address the increasing challenges of managing the nation's soil and water resources, a recommended approach will, obviously, be to review, restructure and consolidate the multiple agencies already existing.

Within the last two decades, there has been some coordinated effort to review and consolidate the laws, agencies and their activities towards managing the nation's soil and water resources [64,65]. A National Water Resources Master Plan was developed in 1995 through a collaborative project between the Nigerian government through the Federal Ministry of Water Resources and the Japan International Corporation Agency (JICA). With the current water-related challenges in the country and inefficiencies in the associated agencies and legal framework, a second project was initiated in 2011, with the focus on reviewing and updating the 1995 National Water Resources Master Plan. The initiative did produce an updated national water resources master plan, focusing on an integrated approach for managing the nation's water resources for the period from 2014 to 2030 [65]. The updated master plan was formulated to address the strategic issues in water resource management, including the increasing challenges of water demand, irrigation, data management, risks and water quality. In the master plan, it was also noted that the responsibilities for water resources development and management in the country are distributed among several government ministries and agencies including the federal and state ministries of Water Resources, Power, Environment, Transport, Agriculture and Rural Development, Mining and Steel as well as local government agencies [65]. The implication is a duplication and overlapping of functions, and the absence of collaboration and communication between these institutions results in gross inefficiencies in water resource management. Hence, a national integrated water resource management approach including river basin management was recommended. The policy document therefore acknowledges the weakness in the current legal and institutional framework and recommends a consolidated approach to be backed by a new legal instrument proposed under the Water Resources Bill. While the 2013 National Water Resources Master Plan is commendable and points the nation in the right direction towards a sustainable management of the water resources, it is limited by data availability, as the needed database is missing and the plan had to rely on coarse estimates [65]. Therefore, efforts must be made to obtain and create the needed database and as well periodically update the plan accordingly.

Following the 2013 Water Resources Master Plan, the National Water Resources Bill was proposed. The bill which was presented to parliament in 2017 seeks to repeal the Water Resources Act Cap W2 LFN 2004; the River Basin Development Authority Act, Cap R9 LFN 2004; the Nigeria Hydrological Services Agency (Establishment) Act, Cap N110A, LFN, 2004; and the National Water Resources Institute Act, Cap N83 LFN 2004, while establishing a National Council on Water Resources, a Nigeria Water Resources Regulatory Commission, River Basin Development Authorities, a Nigeria Hydrological Services Agency, and a National Water Resources Institute. The bill focuses on establishing a regulatory framework for water resources to ensure equitable and sustainable development, management, use and protection of the nation's surface and groundwater resources to ensure present and future access to clean water. While the bill is presented as a consolidated legal framework for managing the nation's water resources sustainably, it has presented new perspectives on water resource use and management in the country. Emphasis was on the right of access to clean water and an integrated water resource management (IWRM) approach adopting the use of hydrological as opposed to geographical boundaries to ensure sustainability. The concept of holistic water resources protection is also promoted, with the recognition of the "polluter pay's principle" also highlighted.

The bill eliminates private ownership and emphasizes the right to use water in accordance with the provisions of the act, which may be similar in some ways to the Land Use Act of 1978, which is neither harmful nor harmless. Its communication and implementation, however, call for reasoning, mostly in the face of the recent push for resource control at the state level [66]. The court is always the point of call for interpretation to ensure the principle of equity and fairness. There needs, however, to be more clarity about where ownership lies. There is also the need to clarify if the ownership of land

grants owners the right to its soil and water resources. On federal, state or local government level, the question of ownership must be addressed, as the long-lingering contest of resource control by states may be triggered. While it appears that this bill is hinged on the assumption of mineral resource ownership by the federal government, its application to water resources may not be popular as there exists a high level of distrust between the people and government [66]. The bill also vested the right to use and manage water in more than one state together with its bed and banks in the federal government. However, the criteria on what constitute surface and groundwater resources affecting more than one state along with their beds and banks are not defined. On regulatory issues, the bill allows for state regulations on water resources, but such regulations are required to follow federal policies and principles. This conforms to other global practices but leaves questions about the relationship between the state and federal bodies in terms of responsibility, accountability and authority. This must be spelt out and exhaustively discussed at national, state and local levels. With the bill requiring states to take all appropriate legal, economic and social measures to control non-point-source pollution, the instruments of the state are being relied upon. If not clarified, this role is bound to be relegated. The new bill emphasizes the polluter-pays principle, which should be a welcome development in Nigeria, but does the country have the legal framework and political will to implement it? There is still a need to improve on the existing legal framework to enhance the implementation of the bill.

A water resources commission to independently regulate water resources in the country is established in the proposed bill. Its membership, however, needs increased numbers of water and water-related science professionals, e.g., hydrologists, hydrogeologists, water resource engineers, etc. The mandate for by-laws, as well as their availability, future review, updates and validity need to be clear, with a demand for their compliance with international standards. The concept of issuing a water license is introduced but the conditions for such a license must be clearly defined, noting that the right of individual or private access to water cannot be removed without a clear practicable alternative. Clarifying the concept of water licensing through an exhaustive stakeholders' forum may reduce public fear. For instance, a license requirement could be obligatory for commercial purposes while individual access alternatives must be categorically and unambiguously exempt from licensing requirements. The bill further establishes a regulatory commission to facilitate technical assistance through research and development. There need to be, however, discussion and policy statements on its funding. While the proposed bill further re-establishes the Nigerian Hydrological Service Agency with its functions unchanged, it is suggested that the agency collaborates with research groups, relevant agencies, universities, etc. to enhance its technical capacity in terms of data acquisition, processing and interpretation. Maintaining a research budget with an open call grant funding instrument for such research will improve the transparency, accountability and quality of such research. An independent national water research institute is also established by the proposed bill. This institute is seen as a duplication of function, demanding more research and infrastructural funding, which has not been available in the past years. Considering the current funding for education and research and the amount of funding available to the ministry, it will be difficult for the ministry to fund such an institute to bring it into a competitive state. Establishing active collaboration with universities through the established agencies would be a preferred alternative.

With the above review of the existing framework, it is obvious there exist several laws and policies as well as agencies for soil and water resources management in the country. What probably is missing is a scientific database, political will backed up with a consistent commitment and an appropriate legal framework for proper implantation of the existing laws and policies. A look at the acts, policies and regulations shows no mention of the basis of decisions and how these will be improved with input from society. While the issue of political will and government commitment cuts across almost all sectors in Nigeria, and its detailed analysis is beyond the scope of this work, this study has focused on data availability, which could drive effective policy formulation and implementation.

## 5. Research and Project Examples

Effective policy formulation and implementation for managing soil and water resources require a detailed understanding of the physical, geochemical and biological processes controlling the storage, flow and transport of fluids and solute within the soil and hydrological systems. Knowledge of hydraulic systems, subsurface sediments and saturated soils as well as their flow and transport properties is required to understand, predict, and ultimately manage these systems in a sustainable manner. For example, policies targeted at protecting soil and water resources from contamination require understanding the nature of the potential contaminants, their sources, transport processes, and pathways from their source to the soil and drinking water sources to be protected. This requires knowledge of the distribution of relevant parameters, such as soil moisture, porosity, permeability, transport velocity, etc., at a sufficiently high resolution to adequately quantify uncertainties that could impact policy implementations. Such an interaction between science and policy, requiring data input from the scientists to aid policy formulation and implementation, has not been well explored in Nigeria [67]. The needed scientific data are basically lacking as research and projects focusing on quantitative estimates of the parameters needed to manage soil and aquifer resources sustainably have received minimal attention [9,23]. For the proposed data-driven policy approach, the scientific database is necessary; hence, we present selected research examples, which can systematically enhance data availability in the following sections. While the research examples presented were carried out in Germany, with additional references to global examples, the approach, techniques and field implementation could be adapted for experimental application in Nigeria. Emphasis is placed on low-cost sensors and instrumentation, which could be an advantage for researchers, policymakers and practitioners considering low research budgets. Ultimately, these examples are presented to stimulate similar thought patterns and encourage experimental and quantitative research for obtaining the needed scientific data in Nigeria and other sub-Saharan African countries.

### 5.1. Field Investigation Methods—Lessons from Lauswiesen Test Site, Tübingen, Germany

While scientific methods for investigating soil and water resources include laboratory and numerical studies, field estimates of the system parameter of interest are often needed. This requires adapting existing methods and developing new ones for local applications considering site-specific constraints. Having a dedicated experimental test site for testing out methods prior to implementation in new areas has proved useful in field-scale research in Germany and other technologically advanced countries. Such experimental sites serve as a 3-D controlled volume field laboratory for method development and the first field-level estimates of parameters that could be used to inform models for predicting parameter distributions needed for policy purposes. Several such experimental sites exist globally, mostly in developed countries, including the Boise Hydrogeophysical Research Site and the U.S. Geological Survey Toxic Substances Hydrology Program research site, both in the United States [68]. The Lauswiesen hydrogeological test site in Tübingen and the Critical Zone Observatory in Rollesbroich, Germany, have also been established for similar purposes [69]. A brief overview of the Lauswiesen test site is presented below as an example.

Adapting, developing and validating field hydrogeological and geophysical methods for subsurface applications has been extensively carried out at the Lauswiesen Hydrogeological Research Site of the University of Tübingen, Germany. The test site (Figure 2), with an area of approximately 300 m × 300 m, is located in the Neckar Valley East of Tübingen, Southwest Germany. The subsurface geology at the research site consists of Quaternary sediments, approximately at the upper 10 m, which are underlain by an impervious mudstone formation of the Upper Triassic "Bunter Mergel". The Quaternary sediments at the research site consist of a 1–2 m layer of alluvial fines with high sandy silt content. This is underlain by gravels with varying sand, silt and clay contents up to a depth of about 9–10 m. Several geophysical surveys, e.g., seismic, electrical resistivity, electromagnetics and vertical electrical conductivity logging, have been conducted in order to characterize the subsurface heterogeneity at the site. Additionally, several studies, e.g., [29,70,71] and unpublished reports,

have been conducted in order to characterize the hydrogeological variability at this research site (Figure 3). The depth to groundwater is between 3 and 4 m and varies during the seasons, while the aquifer's saturated thickness is approximately 6 m. Groundwater flow follows the hydraulic gradient of about 0.3% and flows from southwest towards the northeast, which may vary by up to 15° during extreme water level changes of the Neckar River [70].

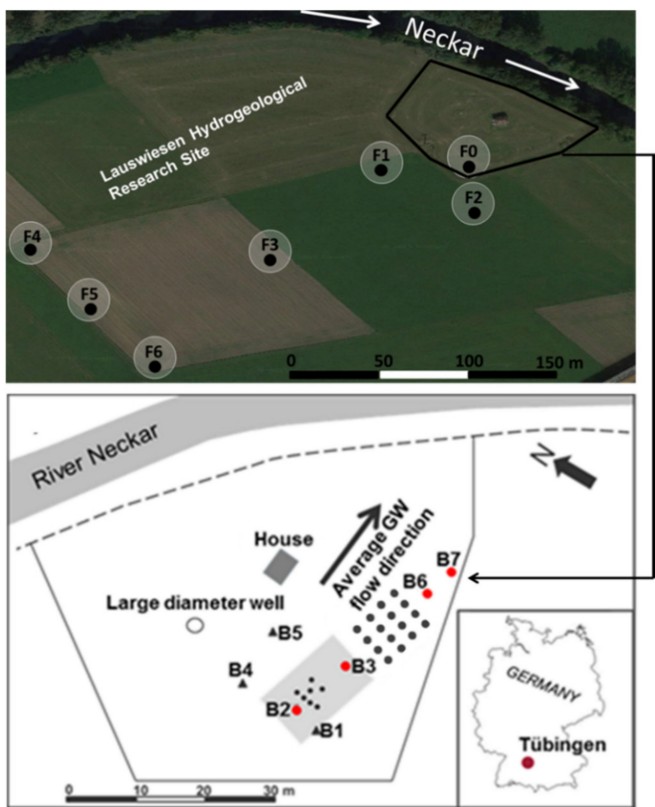

**Figure 2.** The Lauswiesen hydrogeological research site, with wells for groundwater monitoring [72].

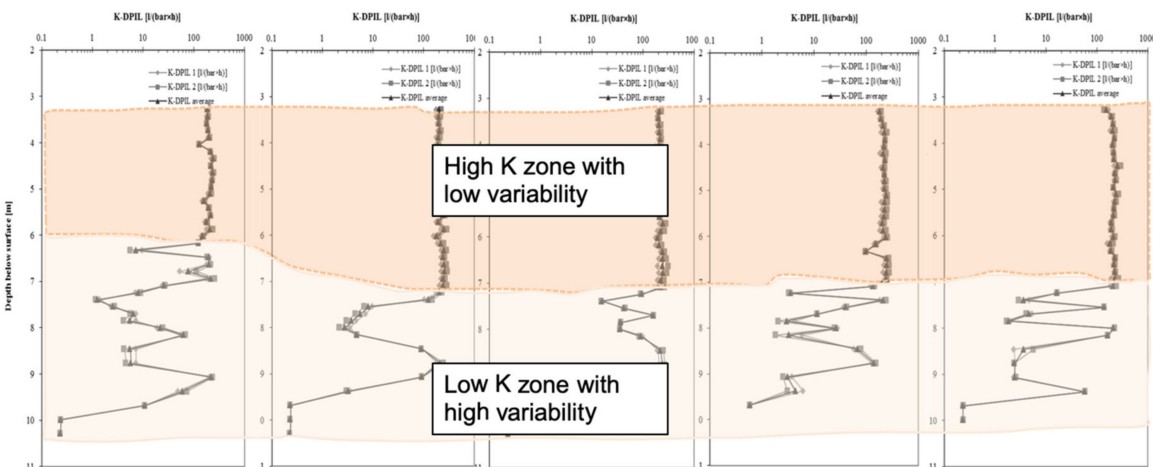

**Figure 3.** Relative hydraulic conductivity distribution at the Lauswiesen Hydrogeological Test Site, measured using the direct push injection-logging technique with data acquired and presented by Uwe Schneidewind and Tao Li in 2008 [71].

With repeated research at the site, the soil and groundwater conditions are relatively well known. Having a field test site such as the Lauwiesen hydrogeological test site allows for improving existing

techniques as well as developing and testing new ones. The site is also used for teaching field courses in hydrology and geophysics, while methods and data from it have informed groundwater management policy decisions in Tübingen and surrounding regions in southern Germany [70]. Setting up a comparable field research site is recommended in Nigeria as none currently exists in the country or is at least not known to the authors at the time of this work. Such a site will allow for the desired technological exchange and adaptation needed to develop experimental techniques and technical competence, as well as the soils and aquifer parameters database needed to effectively manage these resources using a data-driven policy approach.

### 5.2. Low-Cost High-Resolution Techniques for Aquifer Characterization

Field experimental techniques for investigating soil and hydrological systems include conventional and high-resolution methods [72]. Conventional methods, such as pumping [34] and tracer test [70], are mostly limited in giving the needed parameters, such as hydraulic conductivity and specific storage, at a high resolution. While parameter estimates from conventional methods are sufficient for certain policy measures, such as water supply purposes, these approaches are inadequate for policy measures, such as remediation of contaminated soils and groundwater. Such remediation measures require knowledge of soil and aquifer parameters at a high spatial and temporal resolution. Field application of conventional methods [34], though they may require some adaptation, is quite direct. High-resolution methods, in contrast, require high-level instrumentation and data post-processing, which could be expensive. The use of low-cost sensors and instrumentation, however, make them affordable at a low budget. Some selected examples are presented below to provide research impetus in a similar direction.

The development and implementation of soil and hydrological testing in a tomographic sequence allow for estimating their hydraulic properties at a high spatial and temporal resolution. An example is tracer testing in a tomographic sequence, which involves a sequential multi-level tracer injection with tracer breakthroughs observed in each case at different observation wells and depths. An experimental design and field application of tomographic tracer testing was implemented at the Lauswiesen test site, using heat as an artificial tracer (Figure 4). The choice of a thermal tracer is based on the ease of measuring temperature with low-cost 1-Wire sensors [13]. Results of tomographic heat-tracer experiments were in line with earlier work characterizing the aquifer at the test site. The experiments demonstrate that tracer tomography is applicable and suitable at field scale using heat as a tracer. The experimental results also demonstrate the potential of heat-tracer tomography as a cost-effective means for characterizing aquifer heterogeneity at a high resolution.

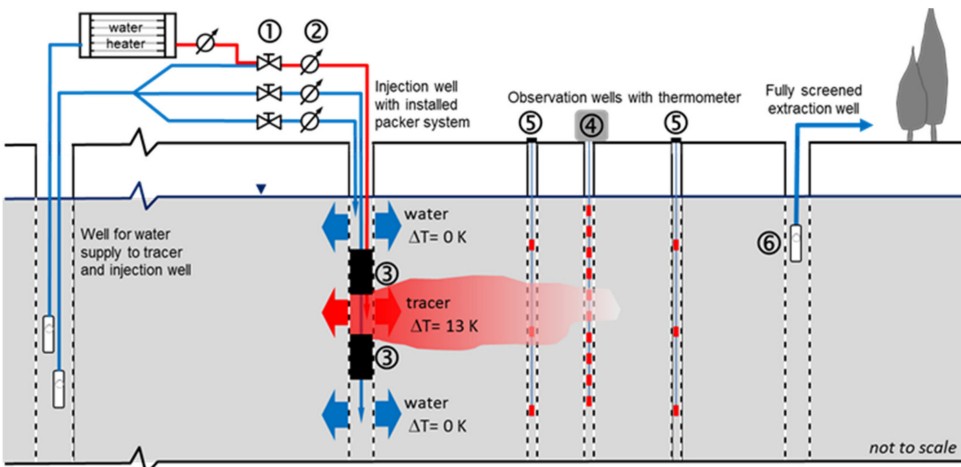

**Figure 4.** Conceptual set-up of the experimental design with a multi-level injection system (1: flow control valves, 2: flow meter and pressure gauges, 3: double-packer system, 4 and 5: observation well with thermometer chain, 6: extraction pump in fully screened pumping well). The red area schematically illustrates the distribution of the heat tracer in the aquifer [38].

This concept is used to demonstrate possible innovative approaches, adapting conventional methods and giving rise to new techniques. Additionally, the use of low-cost temperature sensors enables such implementations even at low research budgets.

### 5.3. Adapting Instrumentations and Methods—4D ERT Tracer Imaging

Often, adapting conventional methods or developing new techniques could imply that existing instrumentation may not be suitable when used in the standard mode. This creates a challenge of instrumental or process adaptability which requires either testing new ways of using equipment or buying modern ones with the needed capabilities, which may have cost implications. Challenging local scientists within Nigeria and other sub-Saharan African countries to explore advancing new ways of applying conventional methods, this is not without challenges requiring outside-the-box thinking. An example of adapting a high-speed measurement configuration of a conventional resistivity meter is presented as a guide. The development and field validation of heat tracer tomography and saline tracer tomography monitored using electrical resistivity (Figure 5) demonstrate the possibility of acquiring multiple hydrogeological and geophysical datasets that could help in improving the challenge of non-uniqueness associated with soil and aquifer parameters estimation [57]. This also confirms that newly developed tomographic techniques could be applied at field scale for simultaneous acquisition of multi-hydrogeophysical datasets.

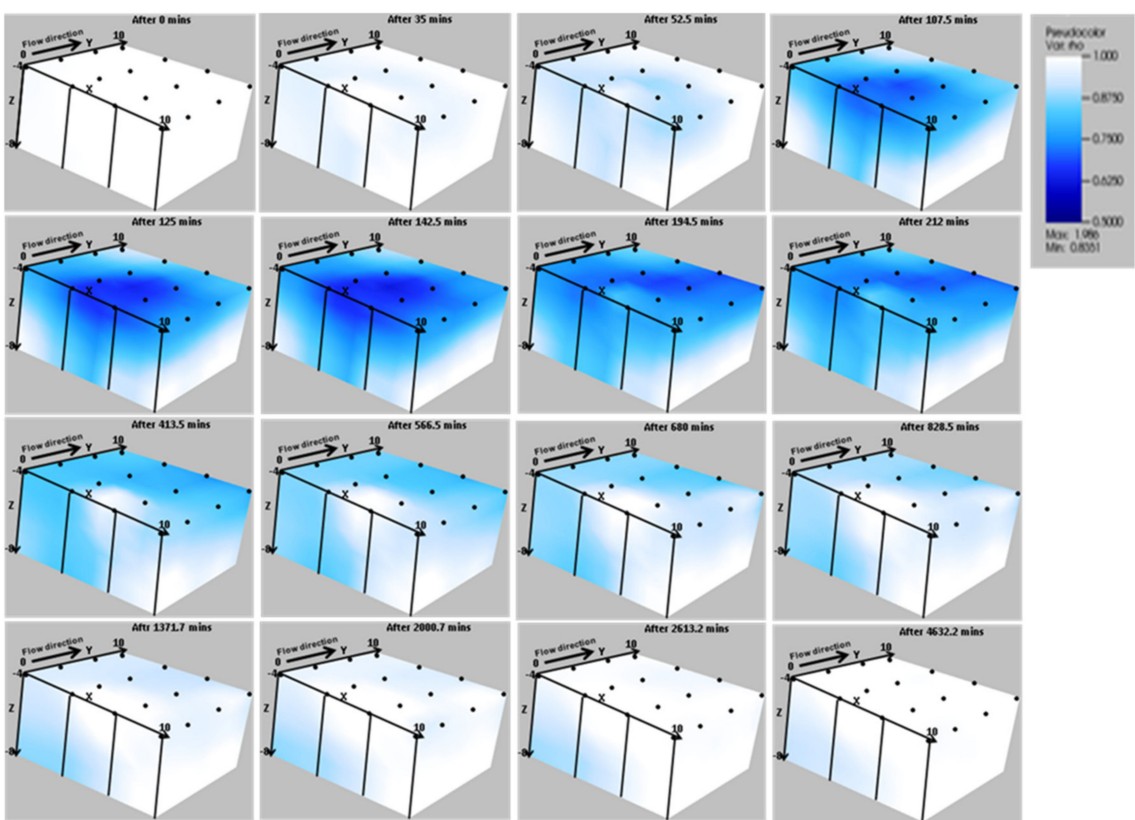

**Figure 5.** Time-lapse 3-D cross-borehole ERT images of resistivity changes reflecting salt tracer transport through the investigated domain [38].

## 6. The Nigerian Challenge and Lessons from Other Countries

With the increasing threats to soil and water resources in Nigeria, the urgent need for pragmatic steps both by government and the populace to protect these resources and better manage them to avoid a looming crisis cannot be overemphasized. While factors such as the social-economic, political, cultural, and geological features unique to Nigeria must be critically evaluated in formulating

and implementing measures to address the challenges, it is worth noting that the challenge of the protection of soil and water resources is not unique to the country. Several other countries in the developed world, including Germany, the United Kingdom, the United States of America and Australia, face similar threats to their soil and water resources and are taking worthwhile steps in addressing these challenges [72–74]. A critical look at examples of how these countries are addressing these challenges could help with useful lessons that could lead to a knowledge exchange and possible adaptation for use locally. Such exchanges and adaptations also have potential for advancing the current state of the art and competence levels on a global scale. Selected soil and water resource management practices that could provide useful lessons for implementation in Nigeria are presented below:

*(1) Delineation and implementation of water protection zones:* One of the ways in which surface and groundwater is protected is through the establishment of water protection zones. Source water protection involves protecting surface water, e.g., rivers and lakes and groundwater sources, e.g., springs and wells, from contamination. The establishment of protection zones involves a zoning with restricted land use based on the travel time of water to the source and the risk of contamination. This usually consists of zones 1, 2, 3A, and 3B (Figure 6), with increasing distance from the water body or drinking water source [75,76].

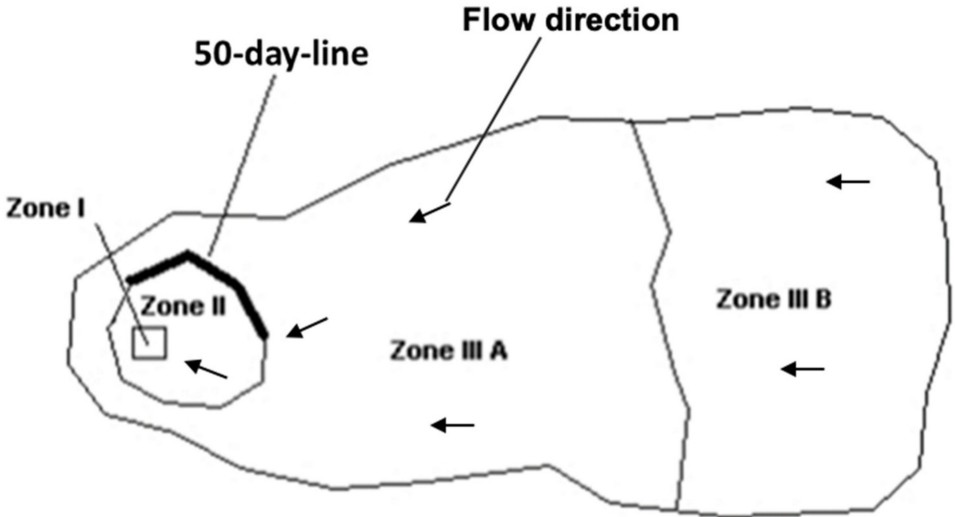

**Figure 6.** Schematics of groundwater protection zone delineation (not drawn to scale).

*Protection Zone 1* refers to the area within the direct vicinity of the drinking water source where water is directly accessible, e.g., a spring or drinking water extraction well. The area is typically cordoned off with a fence and access restricted only to authorized officials. The boundary of protection zone 1 is based on the average time it would take for a biological contaminant to decay and allows for protection against pathogens through the self-cleansing ability of groundwater during its flow and transport. This is established, e.g., in Germany and the United Kingdom, as a 50-day travel time or a minimum radius of 50 m for any contaminant to get to the drinking water source [75,77].

*Protection Zone 2* is the area of high groundwater vulnerability with a delineated area that allows for dilution, reduction of concentration or a significant delay before the potential contaminant arrives at the source. This allows for a decrease in the concentration of the contaminant to an acceptable value or enough time for remediation measures to be taken. Source protection zone 2 is defined by a minimum recharge area to support 25% of the groundwater yield in the protected area, with a minimum radius of 250 m or a 400-day travel time, depending on the local conditions and the rate of abstraction [77]. Additionally, activity restrictions such as prohibiting the application of pesticides or the use of hazardous substances are applied to prevent possible exposure to harmful substances, or, in the case of exposure, an ample reaction time is possible.

*Protection Zone 3* is the entire water source catchment. Restriction on land use practices are required to ensure that hardly degradable hazardous substances are not released into the surroundings.

The delineation of water protection zones follows strict and well outlined guidelines. A detailed experimental and quantitative analysis of flow and transport parameters, including the flow paths, hydraulic conductivity, porosity, velocities, travel times, etc., is a prerequisite for delineating protection zones [75,76]. In addition to a scientific database, one major factor responsible for an effective implementation of the protection zone is its clear communication within the citizenry. This encourages a culture of compliance among the populace.

*(2) Environmental (soil and water) monitoring:* Protecting soil and groundwater resources also requires an established and coordinated monitoring program, involving continuous soil and groundwater sampling and laboratory analysis. Such analyses are then benchmarked against established standards to assess the soil and water health and potential contamination threats. As practiced in Germany, designated municipal authorities regularly monitor the quality of the soil and groundwater within their jurisdictions. When pollutant concentrations in soil and groundwater are found to exceed control standards, necessary measures are taken to investigate the party responsible for such pollution and report it to the central or coordinating competent authority. When concentrations of identified pollutants are below control limits but exceed monitoring limits, regular monitoring is carried out, in most cases with the public and central authority informed. These practices help in managing soil and water resources to ensure access to clean soils and water.

*(3) Obligation to remediate in cases of contamination*: In situations where soil or groundwater contaminations occur, there is a need to for an unequivocal position of the law on whose responsibility it is to remediate the contaminated site. Having and strictly implementing such laws is a prerequisite for ensuring clean soil and water resources. As an example, the German Federal Soil Protection Act (Bundesbodenschutzgesetz) upholds the polluter-cleans principle, where the party responsible for an established contamination is as also responsible for its clean up and remediation. This implies that the authorities can institute a legal process against an identified polluter, which could be the current or former property owner, the occupier or the tenant. This also places the responsibility on property owners with activities that could potentially contaminate the environment, e.g., industries, to embark on regular soil and groundwater monitoring to ascertain their responsibilities in terms of pollution or exonerate them. This also implies that purchaser of properties typically cares about the soil and groundwater health prior to such a purchase, as the purchaser of a contaminated site would generally be liable for its cleanup.

The proactive protection of soil and water resources through proactive actions, such as the delineation and implementation of source water protection zones has been known to reduce the risk of drinking water source contamination in North America, Europe and other countries where it is being practiced [72,78,79]. While these approaches are associated with a significant cost, research across several communities in North America has shown that remediating contaminated drinking water sources could be about 40 times more expensive than the cost of preventing such contamination through source water protection [72].

## 7. A Way Forward—Data-Driven Policy Workflow

Public policy formulation and implementation are complex processes with multiple choices and uncertainties influencing their outcome [80]. The traditional linear approach of reducing policy formulation to the projection of expected outcomes based on a set of possible choices, their probabilities and cost–benefit analysis has failed to give the desired results, mostly in developing countries including Nigeria [80]. Policy experts advocate for new approaches for handling policy complexities, including the integration of scientific data and social elements in the policy formulation and implementation process [80,81]. Policy implementation has been broadly classified into top-down and bottom-up approaches, with the top-down approach starting with a policy decision following down to objectives achieved over time, while the bottom-up approach starts with an operational issue and narrows up to

a policy instrument [82,83]. The approach used so far to address water resources challenges in Nigeria can be described as a poorly implemented "top-down" approach where similar policies are copied from other nations with little or no adaptation for use locally [84]. There is no evidence of rigorous testing, evaluation and adaptation accounting for local peculiarities. Hence, there has been little or no success, despite repeated efforts. Our review of existing policies and their implementation shows that, even with the best intentions of government, these poorly implemented, "top-down", copied policies have been non-sustainable and have little support from the populace [24]. In most cases, as in the case of the proposed 2018 Water Bill, they are perceived by the masses as an extra burden, with little or no effect on alleviating poverty and poor access to these resources [85].

To address the current challenge with soil and water resources in the country, we recommend a data-driven policy model following a mixed "bottom-up" and "top-down" approach [82,83], with bottom-up communication and implementation. This model follows a four-step approach involving policy problem definition; acquisition of the relevant data to better understand the policy problem; analyzing, clarifying and communicating the gathered information; and, lastly, entering the action phase of formulating a policy instrument supported by data. This model, which is similar to a recommended template for healthcare policy formulation [81], focuses on a detailed problem understanding, scenario analysis and public awareness and perception of the problem. For soil and groundwater resource management, which is the focus of this study, we recommend first carrying out a detailed review of the challenges, such as available versus needed drinking water reservoir capacity, farmland irrigation, drinking water source protection, etc. This should be followed by a detailed calibrated hydrological model, incorporating field parameter estimates following experimental examples presented earlier in this study. This should be used to create a public database for evaluating the problem and can be updated over time. Additionally, at this stage, open communication should be established with communities to obtain their perception, understanding and recommendations to develop useful social research elements needed for successful policy formulation and implementation. With improved system understanding, a publicly available database, a predictive model and public input considering vital social elements [62], policy solutions should be formulated with input from all stakeholders. This will help win the buy-in of communities who would become the stewards of such policies. During implementation, a quantitative assessment of policy objectives vis-à-vis achieved results and public participation and perception must be carried out, with feedback used to improve on such policy measures. This recommended framework to managing soil and water resources is summarized in Figure 7. While this approach may not be all-encompassing in terms of policy formulation and implementation (details of which are beyond the scope of this study), it provides a first framework for a sustainable management of soil and water resources in Nigeria and possibly other developing countries.

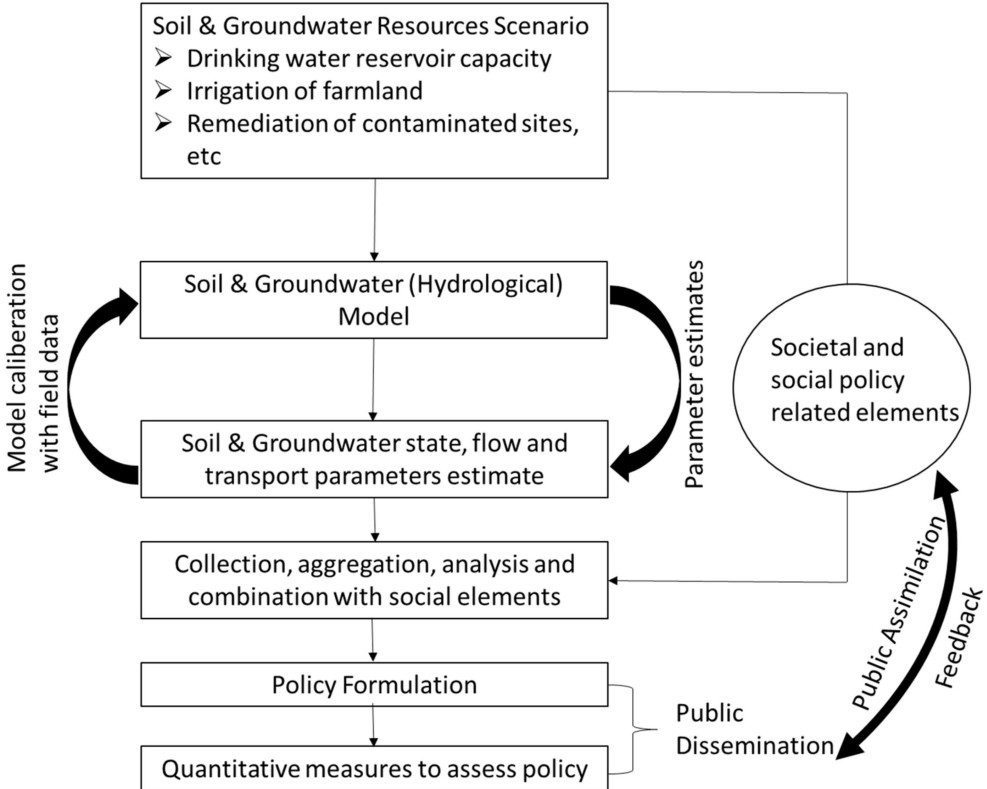

**Figure 7.** Proposed workflow for a data-driven policy implementation for soil and water resource management.

## 8. Conclusions

*Call to the Academic Community*

The volume of research carried out to date on soil and water resources in Nigeria is acknowledged. Despite the challenging research environment characterized by unimaginably difficult constraints ranging from unstable power supply, poor internet facilities and lack of access to current international publications to lack of standard research equipment, some very useful research outputs have been produced, which should serve as a base for creating a database to drive policy in these areas. However, there is a further challenge to scientists working on soil, water and related research areas. There is the need to channel currently available research resources towards experimental and numerical studies that would help provide the needed database for a data-driven policy approach. Most of the policies currently in place in the country, mostly relating to soil and water resources, lack a critical scientific base. While much more is desired from government, it is worth noting that acquiring the right datasets and making them available to the community through an innovative communicative route could stimulate compliance, and may inevitably pressure government to embrace a data-driven approach to policy formulation.

**Author Contributions:** Conceptualization, writing and review, K.O.D. and A.P.A.; Policies and data analysis, K.O.D. and S.E.; Revising intellectual content, K.O.D., S.E. and A.P.A. All authors have read and agreed to the published version of the manuscript.

**Funding:** This research was funded by the American Geophysical Union through the AGU Celebrate 100 centennial grant.

**Acknowledgments:** Some of the research results presented in this work were from studies conducted under the supervision of Carsten Leven and Olaf A. Cirpka, both at the University of Tübingen, Germany, as part of the project "Tomographic Methods in Hydrogeology" in the framework of the GEOTECHNOLOGIEN program supported by

the German Ministry of Education and Research (BMBF). I also acknowledge the Center for International Migration and Development, Germany for supporting my research visit to Nigeria under the Diaspora expert program.

**Conflicts of Interest:** The authors declare no conflicts of interest.

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
