# Peer review of "Sustainable Soil and Water Resources Management in Nigeria: The Need for a Data-Driven Policy Approach"

_sustainability, doi:10.3390/su12104204_

Round 1

Reviewer 1 Report

This is a fascinating and useful paper that should be of interest to academics and policy-makers.

Some comments that I hope will benefit you:

I would get to the point/purpose of the paper earlier; right now, we aren't clear on it until line 84. 

A short introduction to Nigeria would be helpful, starting with a map. A little bit about the geographical and political context in the Intro would clear up inevitable questions about the timing of military rule and other political developments which are mentioned. 

I also wondered about funding the recommended models. Does Germany fund all aspects of water protection or are volunteers used as they are in parts of rural Canada and in Ghana, for instance? 

Might you introduce the term source water protection on page 13?

Re Figure 8, can field data include social research? In some cases the omission of such research means that, say, barriers to water access are not uncovered? Such research could explore the social dimensions of water security, e.g. costs to households, physical disabilities, cultural taboos and water access, etc.

My other points have to do with the writing, which is generally of a high standard:

-- there should be commas around however, 301, 303, 333, etc.

-- some words should be plural but are not singular or the other way around, 307, 457, 551, 578, 600

-- stakeholder's should be possessive plural, 329

-- Low cost should not be capitalized, 361

-- there are problems with a lack of articles, 320, 506, 519, 591

-- the sentence from 427 to 429 needs to be rewritten as it is awkward

-- nothing should be noting, 489

-- shorten the sentence from 487 to 491 or break it in two

-- stirs is not quite the right word

-- the sentence beginning with However needs to be reconstructed for clarity

-- is the use of invariable correct? 600

-- in Figure 8, is assimilation the right word?

Good luck with revisions. I hope that this paper gets the attention of policy-makers in Nigeria and beyond.

Reviewer 2 Report

Major Comments

This is a largely descriptive article calling for data driven approaches for soil and water management in Nigeria. The article accomplishes this by first outlining shortcomings in the current political framework for soil and water management, describing some example approaches of data collection, and providing a policy solution.

My major concerns about this manuscript are (1) it is largely descriptive and (2) lacks any methodology from which its conclusions are drawn. The central thesis is that current policy frameworks in Nigeria are inadequate and there is a lack of data to support soil and water resource management. The authors describe the current and potential policy frameworks and make broad and sometimes unsupported statements about policy effectiveness or ineffectiveness. However, the authors do not provide an analytical framework to base this assessment.

The authors also describe some cases of research that can provide data to inform policy/management. First, these examples are extensively described and take over 5 pages. However, these examples are not directly tied to the central tenant of the manuscript. I recommend revising the section to not provide a highly detailed description of the investigations but to tie the data generated to policy and management outcomes. In short, it is much more important to understand how the implementation of these types of research and data can inform Nigerian soil and water management.

The author’s recommendations for a mixed bottom up/top down model of data driven policy is certainly appealing. However, there is not much in the paper that builds up to or directly supports this recommendation. More time should be spent linking how the components of the proposed workflow address problems identified in the policy review. A methodological approach to the policy analysis would probably help tie these sections together.

Overall, the authors are providing what seem to be much needed and important policy recommendations for soil and water resource management in Nigeria. However, the conclusions and recommendations are weakly tied to the rest of the manuscript. Furthermore, the lack of a detailed analytical approach make it difficult to recommend acceptance of this manuscript. I do think a substantial revision could address these concerns

Specific Comments

Lines 157-246: This section is a fairly extensive overview of soil and water related policy that includes some unsupported statements about policy effectiveness or ineffectiveness and other. Please try to include citations where these claims are made; such as, “The decree did not result in any significant change…” As a U.S.-based reviewer, I do not have substantive knowledge to evaluate that claim, so backing these statements up with literature is important.

Lines 246-251: These lines indicate the issues in the policies outlined in the previous few paragraphs. A table or figure that illustrates the redundancies or inefficiencies would be useful. Kauffman (2015, Fig 3) provides an example that quickly illustrates the complexity and overlaps in water governance structures for a particular basin in the eastern United States. The author does a good job supporting this claim in lines 266-275. Is there a way to move the explanation of function duplication and overlap earlier so it is clear to the reader that this is a supported statement?

Lines 252-255: I have to take by word that the reviewed laws and agencies are driven social-political interest and lack scientific base. Again, I don’t have familiarity with these institutions, but from the previous paragraphs, in particular about the NIHSA, they seem to be setup to at least collect data. Furhermore, based on lines 190-217, the lack of effectiveness of the River Basin Development Authorities seems to be a lack of resources or poor implementation if I am reading correctly. If poor implementation performance underlies soil and water management failures, why would improved data change this? If the thesis of this section is improved data frameworks are needed, I would consider revising the previous sections to emphasize why the lack of data driven approaches have led to underwhelming soil and water management when policies and institutions are implemented.

Line 312-318: As a US-based researcher, this is an interesting statement because it is exactly how it is setup in the US. Non-point source pollution management is “delegated” to state authorities; however, some funding is passed through the federal agencies down to state agencies to facilitate implementation. This model is argued for because it can be more flexible and adaptable to local needs. Although I am not aware if that has been well scrutinized.

Lines 353-483: In addition to the comments detailed in the major comments, I’d like to know why these examples are chosen? These are very specific examples of field experiments generating specific types of data, but it is not clear at all how it is tied to data needs. Provide some context for the types of data already available, and the resolution and quality of the data. Then tie the data needs to the types of field sites that can be used in your case study.

Section “The Nigerian Challenge and Lessons from other countries”: Can you provide detail on the effectiveness or shortcomings of these approaches as implemented in Germany. Why is only Germany used as case examples when the start of the section mentions the UK, US, and AUS are other countries taking steps to address similar challenges?

Fig 7: The scale isn’t really useful here because the distance will change based on local conditions. I would include a time scale that indicates the zones are based on distance travelled over a given time.

Line 551: This is a hanging paragraph, that either needs to be elaborated on or removed.

Kauffman, G.J. Governance, Policy, and Economics of Intergovernmental River Basin Management. Water Resour Manage 29, 5689–5712 (2015). https://doi.org/10.1007/s11269-015-1141-5

Round 2

Reviewer 2 Report

The authors sufficiently addressed major and minor concerns in the manuscript. Overall, the authors have improved the arguments for implementing data driven policy approaches. The manuscript is also improved in grammar and style. I have not further substantive suggestions for improvement.

I do have some remaining concerns about the framework upon which the effectiveness of current policies are evaluated. However, as the authors indicate, the data are not well available and this manuscript is not intended to be a rigorous analysis of past policies (and as this manuscript points to, without data we cannot evaluate management effectiveness). 

I commend the authors for developing a wonderful starting off point for serious policy discussions on the need for data driven policies in the region.